

# Filling Data Gaps in Soil Moisture Monitoring Networks via Integrating Spatio-temporal Contextual Information

Weixuan Wang [1], Yizhuo Meng [2], Zushuai Wei [3, *], Linguang Miao [1], Hui Wang [1], Wen Zhang [1, *]

[1]School of Remote Sensing and Information Engineering, Wuhan University, Wuhan 430079, China
[2]Department of Computer Science, University of Southern California, Los Angeles 90089, USA
[3]School of Artificial Intelligence, Jianghan University, Wuhan 430056, China

*Correspondence to*: Zushuai Wei (weizushuai@whu.edu.cn); Wen Zhang (wen_zhang@whu.edu.cn)

## Abstract

As critical inputs for global climate studies, watershed hydrologic modeling, and satellite soil moisture product validation, in situ soil moisture measurements are frequently compromised by sensor-derived data gaps that disrupt hydrological continuity. To overcome this challenge, we develop ST-GapFill, a novel spatiotemporal reconstruction framework integrating multi-source contextual information through two key innovations: (1) Spatial correlation-guided neighbor selection that identifies optimal auxiliary stations; (2) A long short-term memory (LSTM) network is employed to capture the complex temporal dependencies within the soil moisture time series. Validation on in-situ networks demonstrates that ST-GapFill successfully reconstructs soil moisture dynamics with preserved diurnal-phase fluctuations, achieving 0.91 correlation coefficients with ground truth under low missing-rate conditions (<50%). Comparative analysis reveals the ST-GapFill 's statistically superior performance (RMSE reduction: 27.0% vs IDW, 67.8% vs ARIMA). This method establishes a robust spatiotemporal imputation paradigm for environmental sensor networks, effectively bridging observation gaps to support precision agriculture and climate change impact assessments.

**Key words**. Gap-filling; Long short-term memory (LSTM); Soil Moisture; Deep learning

## 1 Introduction

Soil moisture (SM) plays a critical role as a key factor in land-atmosphere interactions, significantly influencing crop growth and surface evapotranspiration capacity. It is not only an important component of the global water cycle and water balance, but also an important indicator of global climate change. SM monitoring is very important in agriculture, ecology and water resource management (S., Williams et al., 2021; Humphrey, V. et al., 2021). Stable and continuously consistent long time series SM data are critical for global environmental and climate change monitoring. Wireless sensor networks (WSN) have become an important means of acquiring SM for the advantages of small size, low cost, and easy deployment (Dorigo, W. et al., 2021). However, missing data is common in soil moisture wireless sensor network systems. Factors such as power outages, emergency maintenance, and communication failures can lead to data loss from some or all sensors. Such missing not only



affects the effectiveness of real-time monitoring, but also negatively impacts subsequent applications such as change trend analysis and time series prediction. Therefore, accurately filling missing values at specific sites and constructing continuous time-series data has become a key challenge in practical applications (Sadhu, A. et al., 2020; ur Rehman, F. et al., 2018).

Many kinds of gap-filling methods for spatio-temporal data have been proposed, which are mainly categorized into two types: methods based on statistical interpolation and methods based on artificial intelligence. Statistical interpolation methods are based on a strict mathematical model, which assumes that the data are smooth in time and space. It predicts missing values by statistical inference. Statistical gap-filling methods based on temporal information use data from the same sensor at different times to interpolate missing values. This includes mean interpolation and linear spline interpolation (Kreindler, D. M. et al., 2016). Statistical interpolation methods based on spatial information utilize spatial correlation to estimate values by using data from other sensors in the network to fill missing values for a given sensor. This approach includes statistical models such as inverse distance weighting (IDW) (Dhevi, A. S. et al., 2014), autoregressive integrated moving average model (ARIMA) (Shumway, R. H., & Stoffer, D. S., 2017), kriging (Oliver, M. A., & Webster, R., 1990), and k-nearest neighbor interpolation (Troyanskaya, O. et al., 2001). Kuo, P. F. et al. (2021) used a kriging estimator to obtain local weather data. The accuracy of the weather station estimator is slightly lower than that of the local sensor estimator. Yamak, P. T. (2019) compared ARIMA with deep learning models for time series imputation in climate data. ARIMA outperformed LSTM/GRU in scenarios with small gaps (<10% missing rate) and stable seasonal trends. Wang, D. W. et al. (2019) proposed an IDW method based on path selection behavior. After filling gaps on sensor data from an open road network, the new method reduces the absolute bias by 17.62% compared to the results of the traditional IDW method. The matrix completion method can also be used to fill gaps by leveraging the correlation between a sensor and multiple sensors, assuming the data are static. Kortas, M. et al. (2020) proposed a data collector based on energy-aware matrix completion. A three-stage recovery framework based on MC, where all missing data were interpolated using the pooling method. Xie, B. et al. (2020) used four methods—IDW, ordinary kriging, multiple linear regression with residual kriging, and radial basis function neural network with residual kriging—to interpolate large-scale SM in deep soil layers. However, missing values in SM are stochastic and complex, and the generalizability of these methods is limited.

The artificial intelligence-based approach performs spatio-temporal modeling by capturing the complex non-linear relationship between spatio-temporal data and the relevant influencing factors to reconstruct the missing values of the spatio-temporal data. Chhabra, G. et al. (2018) demonstrated SVR's effectiveness in imputing gaps in IoT-based air quality sensor data. SVR achieved lower MAE than k-NN and linear regression by capturing local nonlinear dynamics. Bin, Q et al. (2021) proposed a residual-connected LSTM integrated with gradient boosting (XGBoost) for imputing block-missing data in power load sensors. The ensemble model reduced MSE by 10% compared to standalone LSTM. Zainuddin, A. et al. (2022) reviewed ARIMA, SVR, and ensemble methods for geosensor data. The study highlights that ARIMA is suitable for small gaps with linearity, while SVR/boosting excels in nonlinear and high-dimensional scenarios (e.g., multispectral satellite data). Mao, Y. et al. (2019) proposed a deep neural network multi-view learning approach to reconstruct spatio-temporal data from five perspectives: global spatial, global temporal, local spatial, local temporal, and semantic, to fill in successive missing readings of the sensor.



Rivera-Muñoz et al. (2022) proposed a novel matrix factorization technique (i.e., deep matrix factorization or DMF) via a neural network architecture for estimating missing data in WSNs. Yi, X. et al. (2016) proposed a spatio-temporal multi-view learning-based approach that considers both the temporal correlation between different time-stamped readings in the same time series and the spatial correlation between time series at different locations. This method enables the filling of gaps in geo-aware time-series data sets. Although statistical interpolation and artificial intelligence methods utilize the spatial

characteristics of SM to some extent, SM is influenced by a variety of complex factors, and its temporal changes need to be considered in particular (Zhao, H. et al., 2022). The application of spatio-temporal information and deep learning for filling SM data gaps requires further exploration (Shangguan, Y. et al., 2023).

Recurrent Neural Networks (RNNs) perform well in processing time-series data and can capture the temporal dependencies between data, so researchers have applied them to data-completion tasks (Kim, J. C., & Chung, K., 2022). To address the

problem of RNN forgetting early information when processing long sequences, missing value reconstruction is typically performed by LSTM. LSTM is a variant of RNN that retains information about past events through memory cells (Schmidhuber, J., & Hochreiter, S. et al., 1997). LSTM has been proven to be an effective tool for missing value interpolation on traffic flow sensor data with similar spatio-temporal characteristics to SM (Decorte, T. et al., 2024). Hussain, S. N. et al. (2022) used a hybrid convolutional neural network-long-LSTM for predicting a large number of missing values in a time series dataset of

electricity consumption, achieving better interpolation performance than single models. Kim, J. G. et al. (2023) filled gaps in air temperature using LSTM. Zhang, X., & Zhou, P. (2024) filled gaps in air quality data using multiple LSTM-based transfer depth autoencoders and demonstrated the effectiveness of their model. The advantage of deep learning in capturing spatio-temporal contextual information from sensing networks offers new perspectives and methods for SM missing value reconstruction.

In this study, we propose ST-GapFill, a novel hybrid model that integrates spatio-temporal contextual information for SM gap-filling. Unlike traditional methods that rely solely on spatial interpolation (e.g., IDW) or temporal modeling (e.g., ARIMA), ST-GapFill combines dynamic spatial correlation screening with LSTM-based temporal dependency learning. Specifically, it introduces two key innovations:

- Dynamic Spatial Correlation Selection: A Gaussian model adaptively selects neighboring sites with a high correlation
(threshold >0.85), overcoming the limitations of fixed-radius neighborhood selection in traditional methods (e.g., IDW). This ensures that only sites sharing similar environmental dynamics (e.g., precipitation patterns, soil properties) are incorporated as spatial auxiliary inputs, thereby reducing noise from spatial sensors.

- Exploring the applicability of methods across different missing patterns: Existing studies predominantly focus on single missing scenarios, neglecting the complexity of real-world sensor networks. We rigorously evaluate model performance
under three distinct missing patterns: Completely Random Missing (MCR) caused by transient sensor failures, Missing at Random (MR) due to localized maintenance or damage, and Non-Random Block Missing (NMR) resulting from prolonged sensor outages. Through controlled experiments with missing rates ranging from 5% to 50%, we reveal critical insights into how different missing patterns influence model performance. For instance, spatial interpolation (IDW)



performs better than temporal models (e.g., LSTM) in MR scenarios with consecutive gaps, while ST-GapFill excels in
NMR (block missing) due to its iterative multi-source fusion mechanism. This systematic analysis not only identifies
context-specific strengths of existing methods but also informs the design of adaptive gap-filling strategies tailored to
real-world complexities.

The synergy of these innovations enables ST-GapFill to perform robustly across diverse missing scenarios while maintaining
high accuracy with actual observations. This work advances the field by providing a unified framework for spatio-temporal
data reconstruction, addressing both dynamic spatial dependencies and limitations in existing methodologies for handling
complex missing patterns.

## 2 Methodology

### 2.1 Long short-term memory

Long Short-Term Memory (LSTM) is a variant of Recurrent Neural Networks (RNN). LSTM, proposed in 1997 by
Schmidhuber, J. and Hochreiter, S. (Schmidhuber, J., & Hochreiter, S. et al., 1997), alleviates the problem of gradient vanishing
in RNN models and is the most commonly used neural network for modeling time series in deep learning (Nelson, D. M. et
al., 2017).

LSTM adds a cell state to the RNN, i.e., at each time step, LSTM has three inputs: the current input value $X_t$, the output from
the previous time step $h_{t-1}$, and the cell state from the previous time step $C_{t-1}$. LSTM has two output values: the current output
$h_t$, and the cell state at the current time step $C_t$. Compared with traditional RNN, the neural units of the LSTM are memory
cells with unique memory mechanisms. The structure of the LSTM cell is shown in Fig. 1.

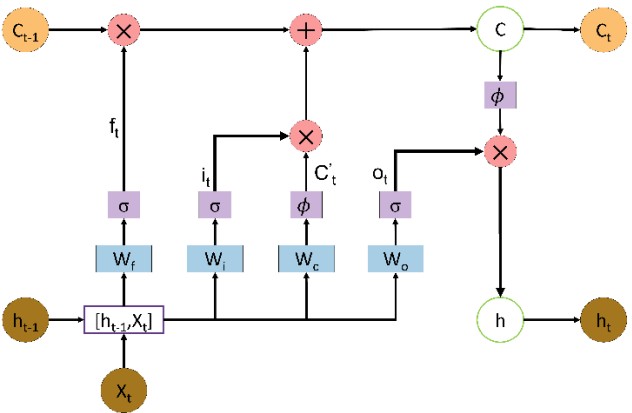

**Figure 1: Structure of the LSTM cell, illustrating the information flow through input gate, forget gate, output gate, and cell state.**

The memory cell in LSTM has three gates: the input GATE, the forget gate and the output gate. The forget gate and the input
gate control the content of the cell state $C_t$. The forget gate determines how much of the previous cell state $C_{t-1}$ is retained,
while the input gate controls how much of the current input $X_t$ is incorporated into the cell state $C_t$. The input $X_t$ and the output





gate control how much of the cell state $C_t$ is output as $h_t$ in the current state. The final output of LSTM is determined by the output gate and the cell gate.

$$f_t = \sigma(W_{fx}x_t + W_{fh}h_{t-1} + b_f), \tag{1}$$

$$i_t = \sigma(W_{ix}x_t + W_{ih}h_{t-1} + b_i), \tag{2}$$

$$C_t' = \sigma(W_{cx}x_t + W_{ch}h_{t-1} + b_c), \tag{3}$$

$$o_t = \sigma(W_{ox}x_t + W_{oh}h_{t-1} + b_o), \tag{4}$$

$$C_t = C_t' \odot i_t + C_{t-1} \odot f_t, \tag{5}$$

$$h_t = \phi(C_t) \odot o_t, \tag{6}$$

$f_t$, $i_t$, $C_t'$, $o_t$, $h_t$ and $C_t$ are the states of forget gates, input gates, candidate cell state, output gates, intermediate outputs and cellular states, respectively; $W_{fx}$, $W_{fh}$, $W_{ix}$, $W_{ih}$, $W_{cx}$, $W_{ch}$, $W_{ox}$, $W_{oh}$ are the matrix weights of the corresponding gates multiplied by the input $x_t$ and the intermediate output $h_{t-1}$ respectively; $b_f$, $b_i$, $b_c$, $b_o$ are the bias terms for the corresponding gates;$\odot$ is the element-wise multiplication of the vector;$\sigma$ is the output of the sigmoid function;$\phi$ is the output of the tanh function.

**2.2 Correlation calculation**

In gap-filling tasks, highly correlated stations enable the model to utilize more relevant information, thereby improving interpolation accuracy. To model the spatial dependence between sites in WSN, a Gaussian model is introduced to calculate the spatial correlation between each pair of sites (Massart, P., & Birgé, L. et al., 2001). For stations with missing data, a correlation threshold is set. Sites with a correlation greater than this threshold are considered more correlated to the target site, and their features are fed into the model for interpolation (Ren, H. et al., 2022).

$$d = 2r \cdot \arcsin\left(\sqrt{\sin^2\left(\frac{\Delta\phi}{2}\right) + \cos(\phi_1)\cos(\phi_2)\sin^2\left(\frac{\Delta\lambda}{2}\right)}\right), \tag{7}$$

$$\text{correlation} = \exp\left(-\frac{d^2}{L^2}\right), \tag{8}$$

d is the distance between the two points; r is the radius of the Earth, typically taken as 6371 kilometers. $\phi_1$ and $\phi_2$ are the latitudes (in radians) of the two points, respectively. $\lambda_1$ and $\lambda_2$ are the longitudes (in radians) of the two points. $\Delta\phi$ is the latitude difference: $(\phi_2 - \phi_1)$. $\Delta\lambda$ is the longitude difference: $(\lambda_2 - \lambda_1)$. d is the distance between the two sites and L is the scale parameter of correlation, where L = 50 km. The scale parameter L=50 km was determined based on the spatial resolution of the SMN-SDR network (grid size of $1°\times1°$) and empirical validation results.



### 2.3 Data pre-processing

Missing values in the original series were replaced with 0 and both SM and rainfall data were normalized before converting data into time-series samples.

$$x_{t,\text{norm}} = \frac{x_t - \mu}{\sigma}, \tag{9}$$

Here, $\mu$ and $\sigma$ denote the sample mean and sample standard deviation of the training data, respectively, ensuring the normalization is consistent with the observed distribution.

The input data for the prediction model includes the target station and its selected neighboring stations, identified based on their strong spatial correlation. The features for each station include soil moisture (SM), precipitation (PP), and a missing value

mask (0 for missing values, 1 for observed values). The target station data is used for prediction, while the neighboring stations' data help provide spatial context. The input to the model at time t can be expressed as the following vector:

$$X_t = [x_t^{target}, p_t^{target}, m_t^{target}, \{x_t^{neighbor(i)}, p_t^{neighbor(i)}\}_{i=1}^n], \tag{10}$$

Where n is the number of selected neighboring stations. $x_t^{target}, p_t^{target}, m_t^{target}$ represent the normalized SM, normalized precipitation and missing value mask at the target station at time t. $x_t^{neighbor(i)}$ and $p_t^{neighbor(i)}$ represent the normalized soil

moisture and precipitation at neighboring stations i, respectively, for time t.

The output, or the target prediction for the target station at time t, is the SM value $y_t^{target}$, which is predicted by the model. Thus, the relationship can be summarized as:

$$y_t^{target} = f(X_t), \tag{11}$$

Where f(·) represents the prediction function of the model (e.g., LSTM).

Eventually, the experimental procedure of this paper is shown in Fig. 2.





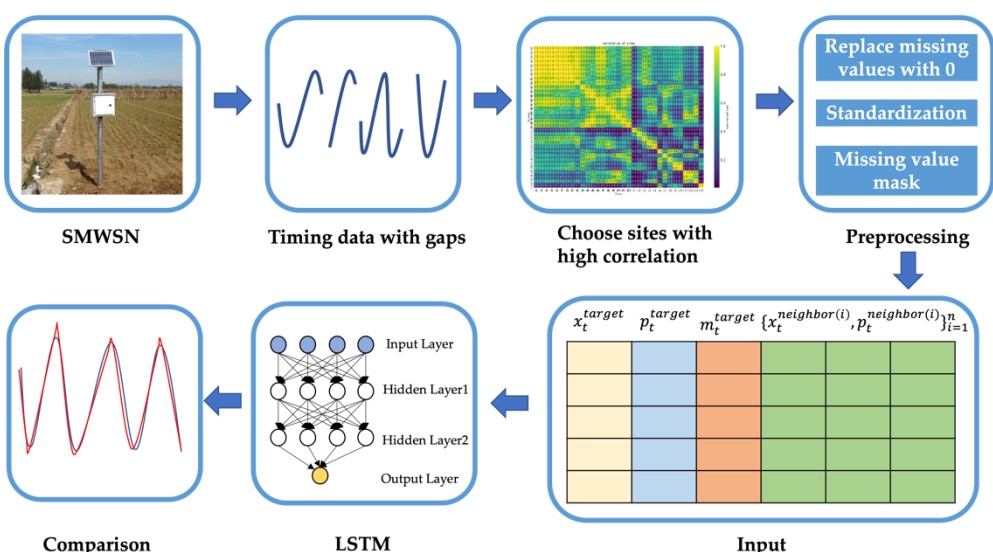

**Figure 2: Experimental flowchart, showing the step-by-step process of the ST-GapFill method for missing data reconstruction.**

## 3 Experiments

### 3.1 Data Description

Shandian River is located in Chicheng County, Hebei Province, at the headwaters of the East Monkey, the birthplace of the Luan River, which flows through the border zone between Hebei and Inner Mongolia. The climate of the basin is a continental monsoon climate, with an average annual precipitation of about 375 millimeters and an annual evaporation of 1188 millimeters. The soils are mainly sand-dune soils, accounting for 50% of the total watershed area, followed by meadow soils, accounting for about 8%. SM varies significantly in the region (Wen, F. P. et al., 2021; Xie, Q. X. et al., 2021). The soils are relatively

dry in spring due to strong monsoon winds and become moist in summer due to abundant precipitation.

In 2018, an integrated remote sensing experiment of water cycle and energy balance was conducted in the Shandian River Basin (Zhao, T. et al., 2020; Yan, G. J. et al., 2021; T. J. Zhao et al., 2021). The experiment is divided into three parts: airborne flight experiment, ground synchronized observation and ground parameter measurement. The data used in this paper are from the Soil Moisture Network Observations (SMN-SDR) in Shandian River Basin. SMN-SDR is a synergistic set of networks for

observing soil temperature (TS), SM and precipitation (PP). It was established during the Luan River Soil Moisture Experiment from July 18, 2018 to September 28, 2018, covering an area of about 10,000 km$^2$ (115.5-116.5°E, 41.5-42.5°N) with a grid size of 1° × 1°. Sensors at three distances of 100 km, 50 km, and 10 km were deployed in the SMN-SDR as shown in Fig. 3. The letters L, M, and S stand for large, medium, and small scales, respectively, and there were 34 stations. The soil moisture sensor used was the Decagon 5TM with measurement depths of 3, 5, 10, 20 and 50 cm. Among the 34 sites, 20 are equipped

with HOBO rain gauges. Power was provided by solar panels and all data were transmitted wirelessly to a server. Data were recorded at 10-minute (pre-June 2019) or 15-minute (post-June 2019) sampling intervals.





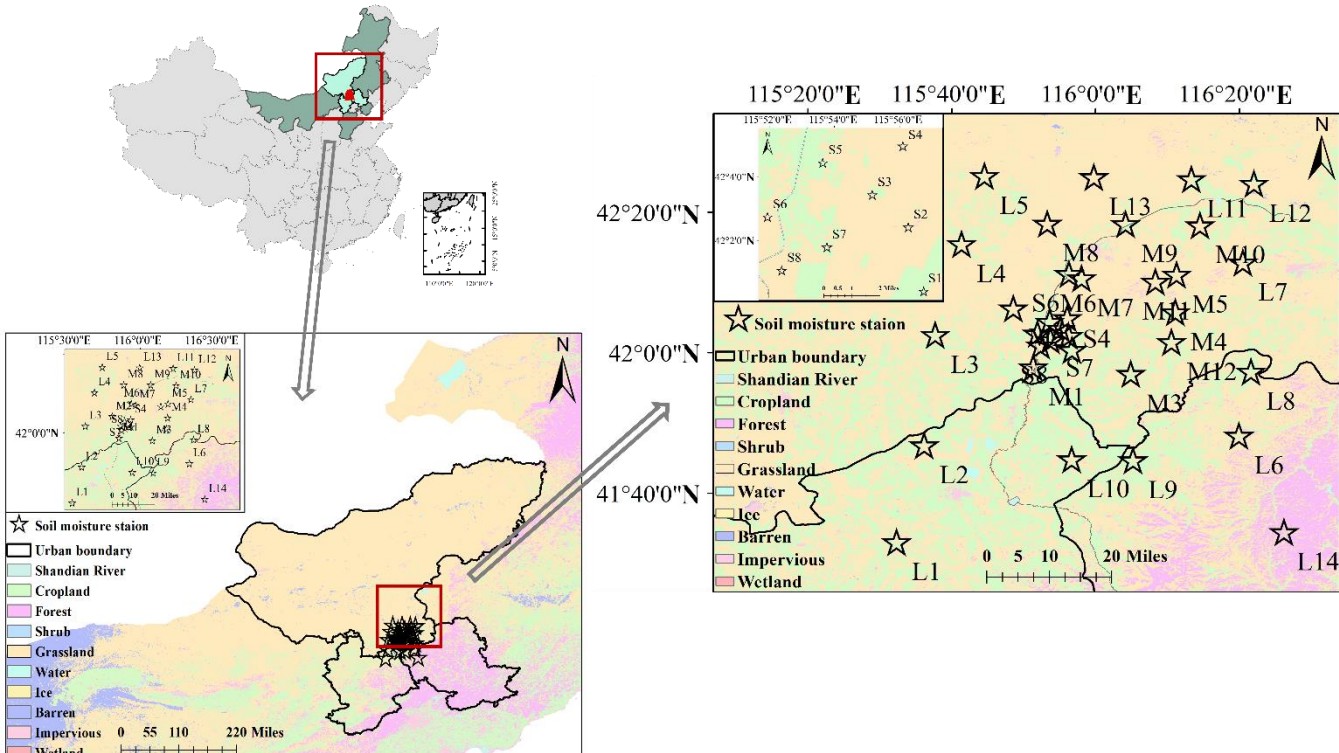

**Figure 3: Schematic diagram of site locations, showing the 34 stations in the SMN-SDR network, with distinct scales for large (L), medium (M), and small (S) sites.**

In this paper, SM and PP at 3 cm depth were used for the experiment. There are only 20 sites with PP. Considering the proximity of precipitation observations from neighboring sites, the geographically closest site was selected as a supplement to the missing rainfall data. There are missing values for up to 16 months in S1, so S1 was excluded from the experiment. All data from the remaining 33 sites were uniformly filtered from 2019.1.1 to 2020.12.31. Raw data were sampled at 10- or 15-minute intervals. A method of taking averages was used to resample all data uniformly for 30 minutes.

**3.2 Baseline algorithms**

Autoregressive Integrated Moving Average Model (ARIMA): ARIMA is based on historical data of time series and fills the gaps by capturing the autocorrelation and randomness of the data. ARIMA transforms the existing non-stationary series into a stationary one through differencing to make it smooth. The autoregressive and moving average components are utilized to estimate the values of the missing data. Interpolation is performed based on the relationship between the previous and 200   subsequent time points and the trend of the error term. Missing data that are consistent with the trend of the original series are generated (Shumway, R. H., & Stoffer, D. S., 2017). The parameters of the ARIMA model were selected through grid search on the validation set with the aim of minimizing the Akaike Information Criterion (AIC). The range of p is 0–6, d is 0–2, and




q is 0–6. Missing values were estimated by iteratively forecasting forward and backward using ARIMA's autoregressive components, conditioned on observed data within the sliding window.

Inverse distance interpolation (IDW): IDW is a method of spatial interpolation based on the distance between a known data point and the point to be interpolated. The core principle is that the closer the known points are to the point being interpolated, the greater their influence, and the farther the known points are, the lesser their influence. The distance between each known point and the point to be interpolated is calculated and the inverse of the distance is used as weights. These weights are later used to perform a weighted average of the values of the known points. Finally, the estimated values of the points to be

interpolated are obtained (Lu, G. Y., & Wong, D. W., 2008). For IDW, the inverse distance weighting power was set to 2, as it provided optimal results in prior soil moisture interpolation studies (Dhevi, 2014).

Support Vector Regression (SVR): SVR models data based on its nonlinear relationships and is used to fill the gaps in time series or spatial data. SVR minimizes the error between the predicted value and the true value by mapping the input data to a high-dimensional space and finding the optimal hyperplane in that space. It also ensures that the model has good generalization

ability. For missing values, SVR fits a smooth interpolation result based on known data points using support vector weights and kernel functions to achieve reasonable prediction and interpolation of missing data (Osman, H. et al., 2021).

### 3.3 Missing Patterns

Referring to the study of Li, L. et al. (2018), the missing patterns of WSN can be categorized into three types: completely random missing (MCR), missing at random (MR), and non-MR (NMR).

1)    In MCR, missing values may occur due to temporary power outages or communication failures. Therefore, they are completely independent. As shown in Fig. 4(a), the missing values are some randomly scattered isolated points.

2)    In MR, missing values may occur due to physical damage or maintenance backlog. Missing values are correlated with their temporally or spatially neighboring readings. As a result, such missing patterns are shown as a number of consecutive points at the same sensor (Fig. 4(b)) or at the same time (Fig. 4(c)).

3)    In NMR, this missing pattern is usually caused by a long-term failure of the sensor and the missing values appear in certain patterns. As shown in Fig. 4(d), the values are missing like blocks.





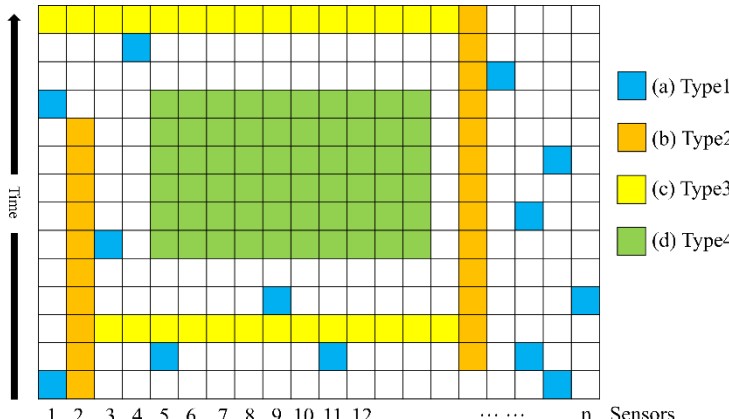

**Figure 4: Patterns of missing values, illustrating the different types of missing data patterns in the SM network.**

## 3.4 Experimental setup

To test the robustness of the proposed interpolation model, four missing data patterns are artificially injected into the dataset. To evaluate the stability of the model under different missing rates, the missing rates of the simulation experiments are set to [0.05, 0.1, 0.15, 0.2, 0.25, 0.3, 0.35, 0.4, 0.45, 0.5]. Sensors and timestamps were randomly selected when generating missing values to mimic the effect of filling in natural data.

To evaluate the model performance, the full dataset was randomly split into training (80%) and testing (20%) sets using

train_test_split from the Scikit-learn library. Within the training set, 20% of the samples were further used for validation through the validation_split parameter of the Keras model fitting function. As a result, the final data allocation consisted of 64% for training, 16% for validation, and 20% for testing. The input size is the total number of features. The batch size of the training data is set to 16 and the model re-peats the training process 50 times on the training dataset to get the best performance of the model. The sliding window size for optimizing the performance of ARIMA, SVR and LSTM was set within the range

of [50,100,150,200,250,300]. The developed LSTM model architecture consists of four layers. The first layer is the Masking layer, which is used to label the "mask values" in the input. The missing value mask is 0 and the non-missing value mask is 1. The middle two layers are two LSTM layers. The last layer is the fully connected layer (Dense), which returns the predicted values for the regression task.

The performance evaluation metrics include Mean Absolute Error (MAE) and Root Mean Square Error (RMSE). The

calculation method is as follows:

$$\text{MAE} = \frac{1}{N} \sum_{i=1}^{n} |(y_i - \hat{y}_i)| , \tag{12}$$

$$\text{RMSE} = \sqrt{\frac{1}{N} \sum_{i=1}^{N} (y_i - \hat{y}_i)^2} , \tag{13}$$





MAE directly calculates the absolute value between the true value and the predicted value. RMSE is the square root of the average squared differences between the true and predicted values, commonly used to assess the deviation between them. The smaller the RMSE and MAE are, the higher the accuracy.


## 4 Results and Discussion

### 4.1 Display of missing data

In this paper, the focus of the experiments is to discuss the effect of interpolation of the model in the presence of a large number of missing values. The model concentrates on modeling pairs of long intervals of consecutive missingness and deals with block

missingness. The SM missing rates for sites that need to be interpolated are shown in Fig. 5. Each site had 35089 pieces of data. More than 10 missing data were considered to be filled in. The number of missing values up to 10 was negligible, and sites with no missing values participated in the assumed experiments after replacing the missing values by taking the mean before and after. L3, L5, L7, L13, M3, M4, and M8 with missing values of 0.03% or more were selected as actual sites to be interpolated. L1, L2, L4, L6, L8, L9, L10, L11, L12, and L14, which have no missing values, are selected as assumed sites.

After introducing missing values for four missing patterns at every assumed sites, simulation experiments were conducted to test the stability of the model at different missing rates.

The selection of L3, L5, L7, L13, M3, M4, and M8 as imputation targets was driven by their significantly higher missing rates (exceeding 0.03%) compared to other stations in the SMN-SDR network, as illustrated in Figure 5. These sites exhibited diverse and severe missing patterns: L7 and M8, for instance, had missing rates of 46.04% and 31.79%, respectively, dominated

by block missing (Type 4) or hybrid patterns (e.g., L7 combined Type 2 and Type 4), which posed substantial challenges for gap-filling algorithms. In contrast, stations like L1 and L2 had near-complete records and were used for controlled experiments where missing values were artificially introduced to validate the model's robustness across simulated scenarios (e.g., MCR, MR, NMR). The chosen targets also represented a strategic mix of spatial scales (large/L, medium/M), ensuring the evaluation covered varied hydrological responses to precipitation and evaporation. Their geographic distribution across the Shandian

River Basin—a region with pronounced SM variability due to monsoonal climate—further allowed the model's performance to be tested under real-world complexities, such as abrupt moisture changes after rainfall or prolonged dry spells. By focusing on these high-missing-rate stations, the study prioritized practical correlation, as their reconstruction is critical for maintaining data continuity in long-term climate and hydrological analyses.

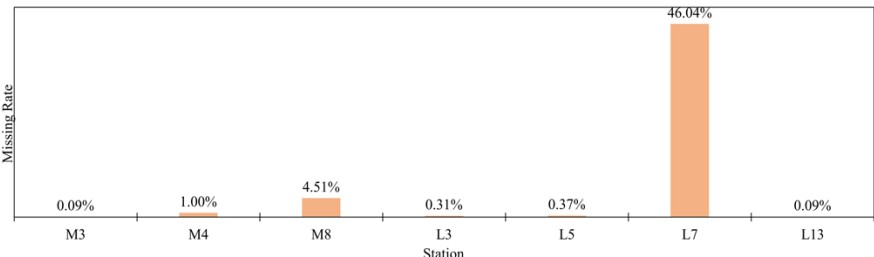



**Figure 5: Missing rates of SM across all stations in the SMN-SDR network, highlighting stations with missing values exceeding 0.03% (e.g., L3, L5, L7).**

Figure 6 shows the missing SM time series. From Fig. 6, it can be seen that there are a large number of single random missing like T1 moments, i.e., MCR. At T2, the values of L7, M4, and M8 are missing, i.e., NMR. At T3, M7 was missing in its entirety, i.e., MR, accompanied by single missing values from other sites. The occurrence of missing values is completely random and unpredictable. Different missing patterns require different models, adding difficulty to the modeling task. So, the relationship between space and time needs to be considered when filling in missing values. When gaps occur at long intervals, it may result in a too small sample size to train the model. Meanwhile, it can be seen from Fig. 6 that the time series of the sites have a high degree of similarity, with similar periods and trends. Therefore, neighboring sites with higher correlation can be used to supplement the lack of features.

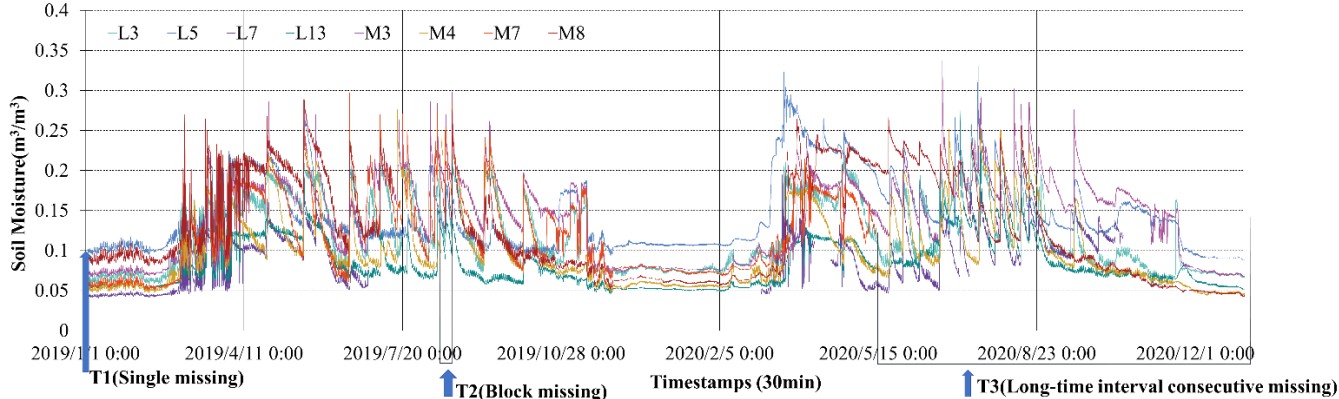

**Figure 6: Missing SM time series under different missing patterns. A visual representation of soil moisture time series data showing the impact of different missing data patterns (MCR, MR, NMR) across multiple stations.**

### 4.2 Neighborhood sites selecting

The correlations between sites are shown in Fig. 7. As can be seen in Fig. 7, most of the correlations for the small-scale sites beginning with S are above 0.7, and some are even close to 1.0. These sites have similar environmental factors and are geographically close to each other. Most of the correlations between the M sites (M1 to M6) are between 0.6 and 0.8. The correlation between sites S (e.g., S5, S6) and M (e.g., M9, M10) is around 0.4 or even lower. Sites L, especially L6, L10, L11, have low correlation with most of the sites (mostly in the range of 0.2 to 0.5). Sites L13, L14 have very low correlation with the other sites, in the range of 0.2 to 0.5. The high correlation sites have similar trends in measured values and are influenced by common environmental factors. The information exchange between sites can make up for the shortcomings of their own features. With a threshold of 0.85, highly correlated sites will be combined as feature inputs to improve the prediction accuracy of the model.





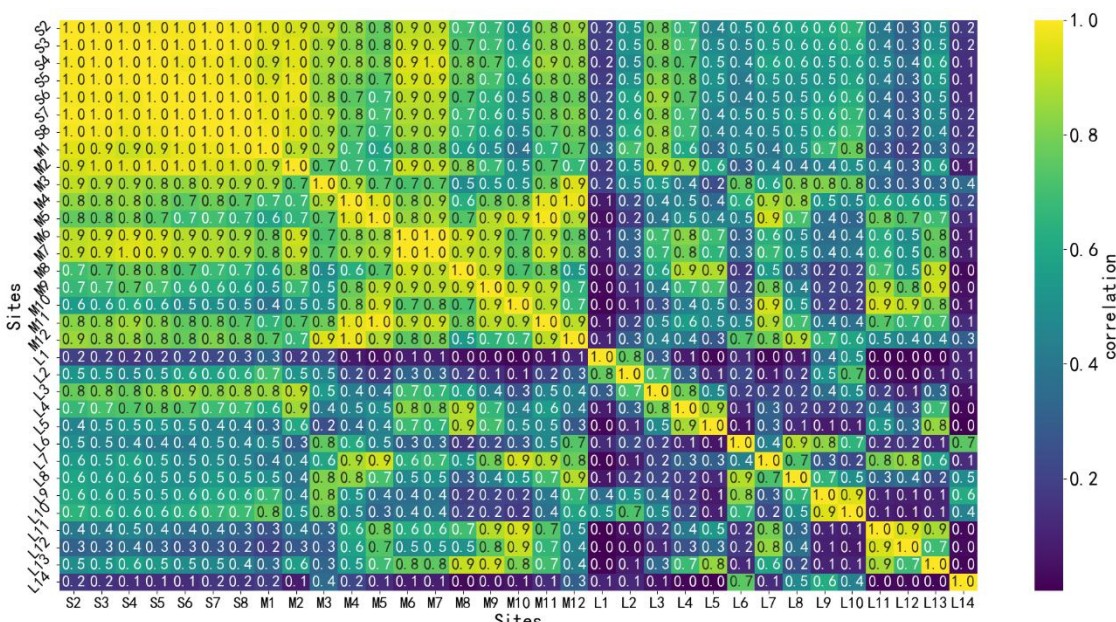

**Figure 7: Correlation between sites, showing high correlations among nearby small-scale sites and varying correlations among larger sites.**

The error shown in Fig. 8 is the comparison of accuracy before and after incorporating neighboring station information. The size of the uniform sliding window is 100, and a comparison test is performed for the actual sites L3, L5, L7, L13, M3, M4, and M8 that need to be interpolated. After adding the correlated sites, most of the sites show lower errors. In particular, L5, L13, M7, and M8 showed significant reductions in MAE and RMSE. L7 showed the highest error, with 14.5% and 0.43% reductions in MAE and RMSE. M7 and M8 showed similar trends, with MAE decreasing by 41.4% and 46.6%, and RMSE decreasing by 61.8% and 20.7%, respectively. L13 showed the most significant change, with MAE and RMSE decreasing by 131.3% and 23.6%, respectively. Some sites, such as L3, L7, and M3, did not have a large difference in MAE or RMSE between the two, but still had a slight advantage with the addition of the correlation site.

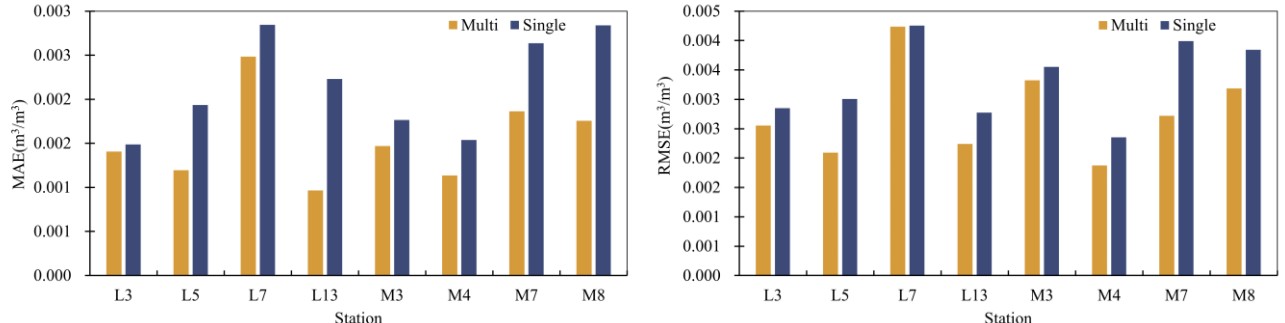

**Figure 8: Comparison of accuracy before and after correlated sites joining, showing high correlations among nearby small-scale sites and varying correlations among larger sites.**





## 4.3 Analysis of simulated data

After individual experiments with sliding window sizes of [50,100,150,200,250,300], the optimal window sizes of 250 for ARIMA, 50 for ST-GapFill, and 50 for SVR were finally obtained, respectively. At the best sliding window size of each model,

L1, L2, L4, L6, L8 with no missing values, L9, L10, L11, L12, and L14 sites were randomly introduced with missing values from the four missing models for a hypothesis experiment.

Figure 9 demonstrates the performance of different models with different missing rates for different missing modes. The trends of MAE and RMSE are the same. For the Type 1 pattern, SVR and ST-GapFill increase the error with increasing missing rate. However, among the four models, ST-GapFill consistently minimizes the error. IDW remained essentially constant, with its

MAE and RMSE consistently around 0.038 $m^3/m^3$ and 0.045 $m^3/m^3$. The error of SVR is greater than that of IDW when the missing rate reaches 40%. For the Type 2 pattern, as the missing rate increases, the long succession of missing values leads to a decrease in the training samples, resulting in fluctuating variations in all four models. IDW consistently maintains the lowest MAE and RMSE. ARIMA has a wide range of fluctuations and is very unstable. For missing rates below 30%, ST-GapFill and SVR perform similarly. After the missing rate is greater than 30%, ST-GapFill is consistently lower than SVR by a small

margin compared to SVR. For the Type 3 pattern, MAE and RMSE of IDW remained near 0.038 $m^3/m^3$ and 0.045 $m^3/m^3$. Although the errors of ST-GapFill and SVR increase with increasing missingness, ST-GapFill consistently performs best and ARIMA performs worst. The error of SVR is higher than that of IDW at a missing rate of 40%. MAE and RMSE of SVR are higher than those of ARIMA at a missing rate of 45%. For the Type 4 pattern, as the missing rate increases, MAE and RMSE of the ST-GapFill are kept at the lowest level, although there is a small increase. The increase decreases after the deletion rate is greater than 15%. MAE and RMSE of SVR are higher than those of IDW at a missing rate of 15%. MAE and RMSE of

IDW remained stable at 0.039 and 0.045 as usual. The $R^2$ values between the actual and predicted values are calculated in Fig. 10. Higher $R^2$ values indicate more consistent estimates. ST-GapFill produces higher correlation values for all missing models, except for IDW, which has the highest correlation for the Type 2 pattern. This indicates that the interpolations generated by ST-GapFill are more consistent with the actual values.



**Figure 9: MAE and RMSE of models for different missing patterns, comparing the performance of various models (ST-GapFill, SVR, ARIMA, IDW) under different missing data patterns.**





| | Missing Rate | ST-GapFill | SVR | IDW | ARIMA | | Missing Rate | ST-GapFill | SVR | IDW | ARIMA |
|---|---|---|---|---|---|---|---|---|---|---|---|
| Type1 | 0.05 | 0.9970 | 0.9992 | 0.9669 | 0.9592 | Type3 | 0.05 | 0.9961 | 0.9930 | 0.8637 | 0.2137 |
| | 0.1 | 0.9935 | 0.9965 | 0.9415 | 0.9252 | | 0.1 | 0.9924 | 0.9878 | 0.8682 | 0.1305 |
| | 0.15 | 0.9878 | 0.9909 | 0.9252 | 0.8669 | | 0.15 | 0.9870 | 0.9807 | 0.8632 | 0.1407 |
| | 0.2 | 0.9837 | 0.9812 | 0.9076 | 0.8570 | | 0.2 | 0.9811 | 0.9701 | 0.8669 | 0.4212 |
| | 0.25 | 0.9751 | 0.9671 | 0.8954 | 0.8156 | | 0.25 | 0.9745 | 0.9583 | 0.8657 | 0.1937 |
| | 0.3 | 0.9655 | 0.9464 | 0.8826 | 0.7756 | | 0.3 | 0.9669 | 0.9421 | 0.8658 | 0.0636 |
| | 0.35 | 0.9485 | 0.9202 | 0.8710 | 0.7424 | | 0.35 | 0.9564 | 0.9290 | 0.8646 | 0.1866 |
| | 0.4 | 0.9448 | 0.8865 | 0.8619 | 0.6881 | | 0.4 | 0.9378 | 0.9090 | 0.8642 | 0.3761 |
| | 0.45 | 0.9322 | 0.8460 | 0.8541 | 0.6532 | | 0.45 | 0.9383 | 0.8941 | 0.8631 | 0.1397 |
| | 0.5 | 0.9105 | 0.8120 | 0.8477 | 0.6279 | | 0.5 | 0.9129 | 0.8889 | 0.8626 | 0.3022 |
| | Missing Rate | ST-GapFill | SVR | IDW | ARIMA | | Missing Rate | ST-GapFill | SVR | IDW | ARIMA |
| Type2 | 0.05 | 0.4518 | 0.3989 | 0.8141 | 0.3201 | Type4 | 0.05 | 0.9790 | 0.9089 | 0.8091 | 0.3338 |
| | 0.1 | 0.3863 | 0.3106 | 0.8083 | 0.2604 | | 0.1 | 0.9778 | 0.8910 | 0.8277 | 0.0624 |
| | 0.15 | 0.4274 | 0.4301 | 0.8263 | 0.2532 | | 0.15 | 0.9533 | 0.8330 | 0.8169 | 0.3964 |
| | 0.2 | 0.4967 | 0.2676 | 0.8164 | 0.2367 | | 0.2 | 0.9370 | 0.7642 | 0.8086 | 0.1646 |
| | 0.25 | 0.3163 | 0.3631 | 0.8351 | 0.2727 | | 0.25 | 0.9396 | 0.7822 | 0.8483 | 0.3827 |
| | 0.3 | 0.3627 | 0.3484 | 0.8713 | 0.2470 | | 0.3 | 0.9357 | 0.7936 | 0.8690 | 0.3457 |
| | 0.35 | 0.3508 | 0.4293 | 0.8172 | 0.2557 | | 0.35 | 0.9384 | 0.7916 | 0.8624 | 0.2112 |
| | 0.4 | 0.4458 | 0.3767 | 0.8362 | 0.2613 | | 0.4 | 0.9292 | 0.8110 | 0.8783 | 0.2903 |
| | 0.45 | 0.2520 | 0.3040 | 0.8458 | 0.2570 | | 0.45 | 0.9224 | 0.8025 | 0.8579 | 0.4013 |
| | 0.5 | 0.4344 | 0.3566 | 0.8426 | 0.2571 | | 0.5 | 0.9101 | 0.8081 | 0.8879 | 0.2855 |

**Figure 10: Comparison of coefficient of determination ($R^2$) under different missing data types. Higher $R^2$ values indicate better**
**agreement between predicted and observed soil moisture.**

From the above analysis, it can be concluded that 1) for Type 2 pattern, the geographic interpolation model (IDW) outperforms
the time series interpolation models (ST-GapFill, SVR and ARIMA). For other types of missing patterns, ST-GapFill performs
best. ARIMA consistently performs poorly because the raw data contain complex nonlinear relationships.2) The time series
interpolation model performs worse than the geographic interpolation model when the missingness rate is high (>40%). 3) For

non-block missing (Type 1, 2, and 3), the interpolation errors of ST-GapFill and SVR become consistently higher with a larger
variation. However, the ST-GapFill error increases less rapidly than SVR. 4) For block missing, ST-GapFill shows a great
advantage. It performs the best across all missing rates, and the error increase rate is minimized. Even with a missing rate of
50%, the model still maintains a low MAE and RMSE.

These results show that ST-GapFill, although a model for modeling time series, the addition of adjacent site information

significantly improves the accuracy of interpolation, especially in the block missing pattern. Neither SVR, which utilizes only
the correlation between timestamps, nor IDW, which utilizes only the correlation between sensors, can adequately account for
data integrity. However, as the missing rate increases, the available information decreases. However, ST-GapFill does not
perform well in all missing patterns. According to the above results, ST-GapFill is not applicable to long interval time series
missing patterns of Type 2, and performs normally under single random missing and simultaneous multi-sensor missing, and

has the best performance under block missing.

In the analysis of simulated data, the ST-GapFill method demonstrated significant advantages over other traditional methods.
For example, compared to spatial interpolation methods based on IDW, ST-GapFill introduces a dynamic spatial correlation
selection mechanism to more accurately identify neighbouring sites with environmental dynamics similar to the target site.
This method not only considers spatial proximity but also incorporates complex nonlinear relationships in time series, enabling

more effective filling of data gaps when handling simulated data with high missing rates. Additionally, compared to ARIMA,





ST-GapFill can better capture long-term dependencies in time series, particularly when handling nonlinear data, where its performance is even more prominent. This indicates that ST-GapFill has greater adaptability and accuracy when processing data with complex spatio-temporal dependencies.

Compared with existing studies, ST-GapFill has obvious advantages in processing data with high missing rates. Chen et al.
(2020) used a spatiotemporal adaptive method, but its accuracy was limited by large-scale continuous missing data. Moreno-Martinez et al. (2020) used a multispectral high-resolution sensor fusion method, but its performance was limited when processing data with high missing rates and blocky missing data. while ST-GapFill dynamically selects relevant sites and uses an LSTM network to capture complex temporal dependencies, making it more effective at handling high-missing-rate block-missing data, with lower MAE and RMSE values.

Under low missing-rate conditions (< 10%), the RMSE of 0.038 m³/m³ approaches the Decagon 5TM sensor's intrinsic accuracy of ±0.03 m³/m³ (manufacturer specification), indicating that the errors introduced by our gap-filling method are no greater than the inherent noise level of the sensor itself, suggesting that the reconstructed data can be considered highly reliable for practical applications. It is noteworthy that the RMSE and MAE values in Section 4.3 (0.03–0.05 m³/m³) are significantly higher than those in Section 4.2 (typically < 0.01 m³/m³). This is expected, as Section 4.2 uses complete observations and
evaluates the effect of incorporating neighboring stations under no artificial gap conditions. In contrast, Section 4.3 involves simulated data gaps under four distinct missing scenarios, where temporal continuity is intentionally disrupted. Especially in NMR (non-missing at random) cases, the target station has no available data during the missing block, forcing the model to rely entirely on external spatial and historical features, which inevitably increases reconstruction error.

## 4.4 Analysis of in-situ data

In order to further verify the upper and lower limits of the ST-GapFill interpolation effect, interpolation was completed for the actual sites L3, L5, L7, L13, M3, M4, and M8 in the original dataset that needed to be interpolated, and the timing variations were plotted as shown in Fig. 11. The missing rate of less than 0.1% was the lowest for M3 and L13, which were all of the Type 4 with block missing. The L3, L5, and M4 missing rates were 0.31%, 0.37%, and 1.00%, respectively. L3 and M4 were Type 4 pattern, and L5 was a Type 2 pattern. From the figure, it can be seen that ST-GapFill can better follow the trend of the
data and is closer to the actual observations when the missing rate is small. ARIMA deviates from the observations. IDW performs smoothly and does not reflect the volatility of the SM well. SVR performs well in some interpolated regions but may be too smooth to capture the abrupt changes and details in the observed data well. The missing rate of M8 is 4.51%, which is a block missing. From the figure, the temporal changes of IDW are articulated too smoothly, and ST-GapFill performs smoothly and is better able to keep in line with the observed data. M7 and L7 have the highest missing rates of 31.79% and
46.04%, respectively, and both are the combinations of the Type 4 missing pattern and the Type 2 missing pattern. ST-GapFill's interpolation results are more consistent with the fluctuating trends in the data, but it performs slightly more conservatively in the time period after August 2020 at site L7, and does not capture the dramatic fluctuations in the observed data. IDW performs well in the long missing time periods, but it does not reflect the fluctuating trends in the data very well. IDW may provide



reasonable estimates in areas where soil moisture is more stable or less variable, but it performs mediocrely in areas of high

fluctuation. The interpolated results of SVR are significantly low, and ARIMA deviates completely from the trend of the actual

observed data.

In the analysis of in-situ data, the ST-GapFill method also demonstrates its unique advantages. Compared with traditional

interpolation methods, ST-GapFill can better reflect the actual trends in soil moisture. For example, when handling data with

low missing rates, ST-GapFill can more closely match actual observational values, indicating its high precision in handling

small-scale data gaps. Furthermore, even in cases of high data missing rates, ST-GapFill can effectively fill data gaps,

outperforming IDW and ARIMA. This indicates that ST-GapFill not only maintains data continuity but also more accurately

reflects the dynamic changes in soil moisture when processing actual observational data.

From a practical application perspective, ST-GapFill offers significant advantages over other methods. For example, Kang et

al. (2019) proposed a data-driven method for filling long-term flux data gaps. Heaton et al. (2019) proposed a method based

on non-Gaussian spatial and spatiotemporal data modelling. Both of these methods are limited by high-missing-rate block-

wise missing data. In contrast, ST-GapFill maintains low MAE and RMSE values when handling block-wise missing data with

high missing rates, indicating its significant advantage in handling complex spatio-temporal data. This superiority arises from

the dynamic spatial correlation selection, which adaptively screens auxiliary stations (Massart and Birgé, 2001), and the

iterative LSTM-based temporal fusion that preserves diurnal variability better than static interpolation or single-model

approaches. These findings demonstrate that integrating multi-source contextual information can bridge longer missing blocks

without sacrificing short-term fluctuation fidelity.

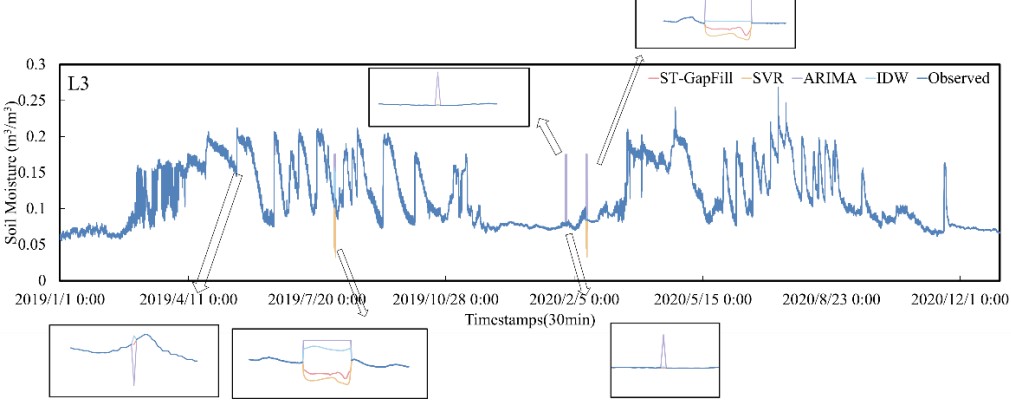



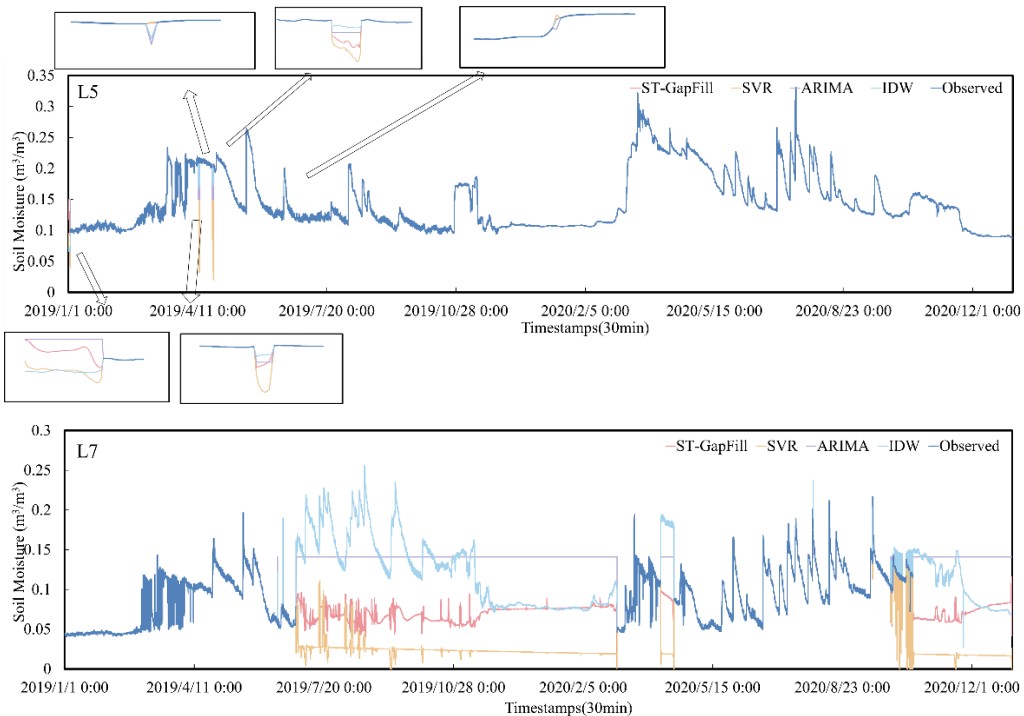


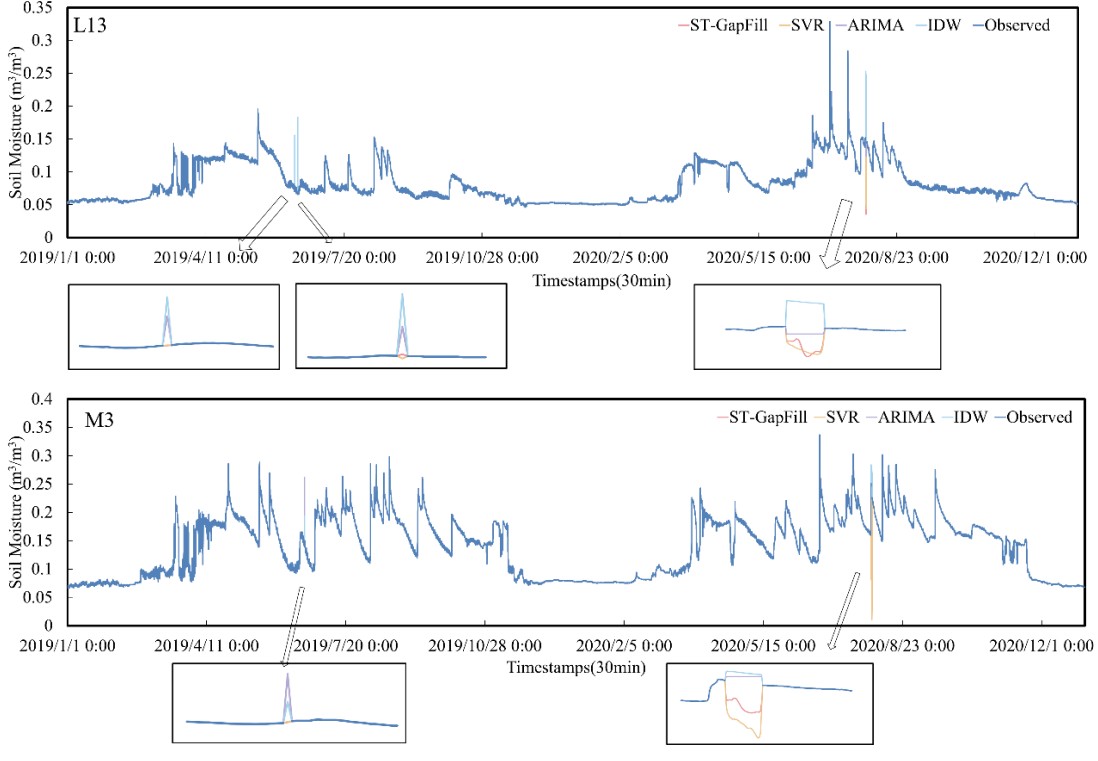



**Figure 11: Timing changes after missing values are interpolated, comparing the results of ST-GapFill, IDW, ARIMA, and SVR for different missing rates.**

## 5 Conclusion

To reconstruct the missing data from the SM automatic monitoring station, this study presents ST-GapFill, the first hybrid model to jointly leverage adaptive spatial correlation screening and LSTM-based temporal modeling for soil moisture gap filling. Its key contributions are:

- Dynamic Spatial Correlation: By replacing fixed-distance neighborhood selection (e.g., IDW) with a Gaussian correlation model, ST-GapFill prioritizes environmentally similar sites, reducing noise from irrelevant sensors. This is particularly effective in heterogeneous landscapes (e.g., Shandian River Basin with mixed soil types).



• Iterative Multi-source Learning: The rolling window framework allows iterative prediction of long missing blocks (up to 50% missing rate) by fusing temporal SM trends, precipitation signals, and spatial correlations. This outperforms static matrix completion methods (Rivera-Munoz et al., 2022) that assume low-rank data structures.

• Superiority in Block Missing: The experimental results show that when the missing rate reaches 40%, ST-GapFill still maintains a low MAE and RMSE of 0.038 $m^3/m^3$ and 0.045 $m^3/m^3$, respectively. For block missing (Type 4), ST-GapFill

significantly outperforms the other models, being able to maintain low MAE and RMSE with missing rates as high as 50%. In contrast, traditional interpolation methods and other machine learning models perform less effectively in the case of high missing rates.

In the validation of real data, ST-GapFill demonstrates good performance. Especially when the missing rate is low, ST-GapFill can effectively capture the dynamic changes of SM, which are highly consistent with the actual observations. For example, the

missing rates of L3 and L5 are low, and the interpolation results match closely with the actual data. And at sites M7 and L8, where the missing rate is high, ST-GapFill outperforms the traditional IDW and ARIMA models, even though it fails to fully capture the sudden fluctuations. This study not only confirms the effectiveness of ST-GapFill in missing time series data filling, but also provides an important theoretical basis for the development of future SM monitoring techniques.

## 6 Acknowledgments

This research was supported by the National Natural Science Foundation of China (grant No. 42301441), and the Research Fund of Jianghan University (grant No. 2023JCYJ13).

## 7 Author Contribution

W.W.: Writing – original draft, Validation, Software, Methodology, Investigation, Formal analysis, Conceptualization. Y.M.: Supervision. Z.W.: Validation, Supervision, Project administration, Methodology, Investigation, Funding acquisition. L.M.:

Methodology, Conceptualization. H.W.: Conceptualization.

**W.Z.: Conceptualization.Competing interests**

The authors declare that they have no known competing financial interests or personal relationships that could have appeared to influence the work reported in this paper.

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
