# Peer review of "Filling Data Gaps in Soil Moisture Monitoring Networks via Integrating Spatio-temporal Contextual Information"

_EGUsphere, 2025_

## Author Comment (AC1)

**Author's Response to RC1**

Dear Editor and Referee:

We are particularly grateful for your careful reading, and for giving us many constructive comments on this work! Those comments are all valuable and very helpful for revising and improving our paper, as well as the important guiding significance to our research. We have studied the comments carefully and have made corrections which we hope meet with approval.

According to the comments and suggestions, we have tried our best to improve the previous manuscript **EGUSPHERE-2025-1900** ("Filling Data Gaps in Soil Moisture Monitoring Networks via Integrating Spatio-temporal Contextual Information"). Here is a summary of the major changes to the new manuscript, and then answer the reviewer's questions one by one.

a) We have revised the overall organization of the paper to make the presentation clearer and more coherent. The literature review section has been expanded and refined to better position our work in the context of existing studies.

b) We have included LOCF as an additional simple baseline method and compared its performance with our proposed approach.

c) To enhance reproducibility, we have created an open repository containing the source code and dataset used in our experiments. The repository is available at "https://github.com/siaahwang/FillingGaps" and the code can be run directly to reproduce the results presented in the paper.

We hope these revisions address your concerns and contribute to improving the quality and reproducibility of our work.

Best regards,

Zushuai Wei and all co-authors

**Author's Response to the Comments of Referee #1**

**Question 1.1**

**Q1.1:**

line 35: "statistical interpolation and methods based on artificial intelligence."

- I think it makes sense to consider the classification in more detail, because this division is quite general, and I did not see any further justification for this particular separation in the text. For example, further in the text, "The artificial intelligence-based approach performs spatio-temporal modeling by capturing the complex non-linear relationship" refers to the specificity of the category "artificial intelligence methods"—nonlinearity. However, k-nearest neighbor interpolation, which the authors included in the first category, allows modeling nonlinear relationships. Firstly, I believe that the division into "statistical interpolation and methods based on artificial intelligence" is unnecessary in the context of this article. Secondly, I suggest reworking this section and providing a more comprehensive and detailed analysis of the solutions. For the sake of systematization, it would be useful to create a table comparing the methods.

**Response:** Thank you for your detailed suggestions regarding the classification and analysis section. We have comprehensively revised the classification and overview of interpolation methods in the introduction:

➢ Redesigned classification approach: Replaced the original broad dichotomy of "statistical interpolation methods vs. artificial intelligence methods" with a more rational four-category framework: physics-based methods, traditional statistical/deterministic interpolation methods, machine learning methods, and deep learning methods. This better reflects the modeling principles and applicable scenarios of different methods.

➢ Enhanced explanations for each category: Representative algorithms (e.g., IDW, Kriging, ARIMA, SVR, LSTM) have been added for each category, detailing their respective advantages, disadvantages, and applicability conditions. Particular emphasis is placed on their linear versus nonlinear modeling capabilities and differences in modeling spatio-temporal dependencies.

➢ Add a comparative table: A new systematic table has been introduced to contrast the differences between methods across dimensions such as "consideration of spatio-temporal characteristics,

nonlinear modeling capability, adaptability to high missing rates, computational complexity, real-time performance, and strengths/limitations." This facilitates readers' quick understanding of each method's characteristics.

**Table 1 Performance Comparison of Different Methods.**

| Method | Spatiotemporal characteristics? | Nonlinear modeling? | Adaptation to high loss rates | Computational complexity | Real-time capability | Advantages | Limitations |
|--------|-------------------------------|---------------------|------------------------------|-------------------------|---------------------|-----------|-------------|
| Physical Model | Yes | Partly | High | High | Low | Highly physically consistent. | Relies on external models and driving data. |
| IDW | No | No | Low | Low | High | Suitable for scenarios with dense data and small areas. | Highly heterogeneous regions exhibit large errors. |
| Kriging | No | No | Low | Medium | General | Theoretically optimal linear unbiased estimator. | Not applicable to non-stationary/nonlinear fields. |
| ARIMA | No | No | Low | Low | High | Skilled at identifying trends and seasonal patterns. | Long-term prediction error accumulation. |
| LOCF | No | No | Low | Low | Low | The algorithm is simple. | Prone to introducing systemic bias. |
| SVR | No | Yes | Medium | Medium | General | Strong learning ability. | Depends on training data and parameter tuning. |
| LSTM | Yes | Yes | High | High | General | Skilled at identifying long-term dependencies and dynamic changes. | The physical explanation is difficult. |

These modifications not only address your requests for classification rationality and systematicity but also enhance the comprehensiveness and readability of the paper's methods review section.

**Question 1.2**

**Q1.2:** Line 110: Methodology section

- I suggest changing the order of the section: first, discuss Data pre-processing, then Correlation calculation, and then Long short-term memory. This will make the narrative more coherent: from data to the correlation analysis method, and then conclude with an explanation of the final model. At the same time, I suggest paying special attention here to explaining why this particular architecture of the final algorithm was proposed (clearly specify in the text what each individual block

is responsible for), while the explanation of how the LSTM architecture of neural networks works is not an important part of the narrative. It will be enough to provide a link and not focus on this.

**Response:** Thank you for your valuable suggestions. We have revised the Methods section as follows based on your feedback:

➢ Adjusted chapter order: Placed "Data Preprocessing" first, followed by "Correlation Calculation," and concluded with "LSTM Model." This sequence ensures a logical progression from data preparation and neighboring station selection to final model construction, enhancing overall coherence.

➢ Simplified LSTM principle description: Removed detailed derivations of LSTM gating structures and formulas, retaining only a brief background introduction with reference links for further reader exploration.

➢ Strengthen motivation for architecture design and module functionality: Emphasize the roles of the adopted Masking layer, double-layer LSTM, and Dense layer, along with how the sliding window design aids in capturing temporal dependencies. Further clarify how each module collaborates to address missing data reconstruction and why this architecture suits modeling the spatio-temporal characteristics of soil moisture.

**Question 1.3**

**Q1.3:** Line 235: To evaluate the model performance, the full dataset was randomly split into training (80%) and testing (20%) sets using train_test_split from the Scikit-learn library.

- This can be kept as it is, but I think not every researcher in the field of geosciences knows Python programming so it might be useful to explain this step in plain text. And since we are talking about using Python here, I would be happy to take a look at the source code of the experiments. I suggest that the authors make their model available as an open-source solution and create a repository on GitHub (if it is legally possible).

**Response:** Thank you for your valuable suggestions. We have made the following improvements to this section:

➢ Added explanatory text: When describing "using train_test_split to randomly split the dataset into training and testing sets," we included a plain-language explanation: All samples are

randomly divided into two parts, with 80% used for training the model and 20% for evaluating its performance.

➤ Open-source code: To facilitate research replication, we have organized and open-sourced the experimental code and model training scripts for this study on GitHub. The link is as follows: https://github.com/siaahwang/FillingGaps.

**Question 1.4**

**Q1.4:** Line 250: The smaller the RMSE and MAE are, the higher the accuracy

- It is better to avoid using the term "accuracy," which has a very specific meaning in machine learning. It is better to say "the smaller the error, the better the model."

**Response:** Thank you for your thoughtful suggestion. We have revised the original sentence "**The smaller the RMSE and MAE are, the higher the accuracy**" to "**The smaller the RMSE and MAE are, the better the model performance**".

**Question 1.5**

**Q1.5:** Line 300: Figure 7: Correlation between sites, showing high correlations among nearby small-scale sites and varying correlations among larger sites.

- Since the article discusses "Spatio-temporal Contextual Information," it would be useful to include a map (there is space on the right) to show the location of these stations. You can even take a specific station and show its neighboring stations in color depending on the correlation coefficient used: dark blue dots if the correlation coefficient is weaker, and yellow if it is stronger, just like on the matrix.

**Response:** Thank you for your valuable suggestions. We have updated Figure 7 as per your recommendations:

➤ Added geographical location map: A spatial distribution map of observation stations has been added to the right of the correlation matrix to visually display the actual locations of stations within the watershed.

➤ Color-mapped correlation coefficients: Using L11 as an example, we visualized its correlation

coefficients with other stations using a color gradient. Sites with higher correlations appear closer to yellow, while those with lower correlations appear in dark blue, consistent with the correlation matrix on the left.

[Figure]

**Question 1.6**

**Q1.6:** Line 340: Figure 10: Comparison of coefficient of determination ($R^2$) under different missing data types. Higher $R^2$ values indicate better agreement between predicted and observed soil moisture.

- Please give an example of how $R^2$ is calculated; the formulas for MAE and RMSE, for example, are given above. It may be useful to calculate adjusted $R^2$ and visualize it, instead of the regular coefficient of determination.

**Response:** Thank you for your suggestion. We have added the formula for calculating $R^2$ (coefficient of determination) in the Methodology section to help readers better understand the meaning and calculation of this metric. Additionally, we have clarified in the main text why we selected the standard $R^2$ rather than the adjusted $R^2$: Since our primary focus is comparing the reconstruction performance of different models on the same dataset, $R^2$ sufficiently reflects the goodness of fit. Therefore, we did not compute the adjusted $R^2$ separately. We have retained the original $R^2$ results in Figure 10 to maintain consistency with other performance evaluation metrics.

**Question 1.7**

**Q1.7:** Line 420: 5 Conclusion, Figure 11

- Thank you for the clear visualization of the modeling results. Looking at the graphs, it seems to me

that simple methods, such as LOCF (last observation carried forward) method or, if the experimental setup allows, linear interpolation of the time series, could perform just as well as the baseline approaches considered here (for example ARIMA or SVR). I understand that predictive models based on previous values cannot "look" into the future. However, from the problem statement, I do not see any restrictions on why information before and after the gap cannot be used to fill it. In any case, could you please add LOCF here for comparison.

**Response:** Thank you for your suggestion. We have supplemented the Methods section with an introduction to the LOCF method and added comparative results for LOCF in the Experimental section (Figures 9 and 11). Analysis indicates that LOCF provides reconstruction accuracy comparable to ARIMA and SVR when missing rates are low. However, as missing rates increase or when long-term gaps occur, errors significantly rise, rendering LOCF incapable of capturing dynamic changes in soil moisture. These findings further demonstrate the superiority and necessity of the ST-GapFill method proposed in this paper for addressing complex missing data patterns.

[Figure]

**Figure 9: MAE and RMSE of models for different missing patterns, comparing the performance of various models (ST-GapFill, SVR, ARIMA, IDW, LOCF) under different missing data patterns.**

[Figure]

[Figure]

**Figure 11: Timing changes after missing values are interpolated, comparing the results of ST-GapFill, IDW, ARIMA, LOCF and SVR for different missing rates.**

**Question 1.8**

**Q1.8:** Line 190: Figure 3

- The markings (dots) are hard to see. You could use regular dots instead of "stars". And also you could show the L, M and S with color to make it easier to distinguish.

**Response:** Thanks for this comment. Here is the revised image.

[Figure]

**Question 1.9**

**Q1.9:** Line 230: Figure 4

- Please add names to types 1-4 in the image, so it will be easier to connect the visualization to the text. For example "(a) Type1 - completely random missing", etc.

**Response:** Thanks for this comment. Here is the revised image.

[Figure]

**Question 1.10**

**Q1.10:** Line 245: The performance evaluation metrics include Mean Absolute Error (MAE) and Root Mean Square Error (RMSE). The calculation method is as follows: ...

- Please indicate what y means in the formulas and what y with a hat means. I know most readers will understand, but it is better to be clear.

**Response:** Thank you for your suggestion. We have added clarification after the MAE and RMSE formulas to explicitly state that $y_i$ denotes the observed value, $\hat{y}_i$ denotes the model's predicted value, $n$ denotes the total number of samples, $\bar{y}$ denotes the mean of the sample.

**Question 1.11**

**Q1.11:** Line 285: Figure 6

- It may make sense to move the footnotes with gap indicators to the top so that they do not overlap with the dates on the X-axis.

**Response:** Thanks for this comment. Here is the revised image.

[Figure]

**Question 1.12**

**Q1.12:** Line 340: Figure 10: Comparison of coefficient of determination (R2) under different missing data types. Higher R2 values indicate better agreement between predicted and observed soil moisture.

- In this context, I suggest using "better consistency" instead of "better agreement"

**Response:** Thank you for your suggestion. We have revised the caption of Figure 10 from "better agreement" to "better consistency" to avoid ambiguity and more accurately convey the alignment between the predicted results and the observed values.